# Genome-Wide Identification and Expression Analysis of the *Dof* Transcription Factor in Annual Alfalfa *Medicago polymorpha*

**DOI:** 10.3390/plants12091831

**Published:** 2023-04-29

**Authors:** Linghua Yang, Xueyang Min, Zhenwu Wei, Nana Liu, Jiaqing Li, Youxin Zhang, Yuwei Yang

**Affiliations:** 1College of Animal Science and Technology, Yangzhou University, Yangzhou 225009, China; 2Institution of Grassland Science, Yangzhou University, Yangzhou 225009, China

**Keywords:** *Medicago polymorpha*, *Dof* transcription factor, bioinformatics, abiotic stresses, gene expression

## Abstract

The *Dof* transcription factor is a plant-specific transcription gene family that plays various biological functions in plant development and stress response. However, no relevant research has been conducted on *Medicago polymorpha*. Here, 36 *MpDof* genes were identified in the *M. polymorpha* genome and further divided into 10 groups based on the comparative phylogenetic analysis. The essential information of *MpDof* genes, such as chromosomal localization, gene structure, conserved motifs, and selective pressures were systematically analyzed. All 36 *MpDof* genes were predicted to contain more *cis*-acting elements related to hormone response. *MpDof24* and *MpDof25* were predicted to interact with *MpDof11* and *MpDof26* to involve in the photoperiod blooms process. The *MpDof* genes showed a diverse expression pattern in different tissues. Notably, *MpDof29* and *MpDof31* were specifically expressed in the large pod and root, respectively, suggesting their crucial role in the pod and root development. qRT-PCR analysis indicated that the expression levels of *MpDof10*, *MpDof25*, *MpDof26*, and *MpDof29* were obviously up-regulated under drought, salt, and cold stress. Collectively, genome-wide identification, evolutionary, and expression analysis of the *Dof* transcription gene family in *M. polymorpha* will provide new information to further understand and utilize the function of these *Dof* genes in *Medicago* plants.

## 1. Introduction

Transcription factors (TFs), also known as trans-acting factors, interact with *cis*-acting elements in the promoter region to regulate gene expression [1]. DNA-binding One Zinc Finger (DOF) is a plant-specific TF and plays a crucial role in the process of plant growth and development. The N-terminal of the *Dof* proteins has a highly conserved single zinc finger DNA-binding domain consisting of 50–52 amino acid residues, which can specifically bind to 5′-(T/A) AAAG-3′ sequences in the target gene promoter [2,3]. The C-terminal domain of *Dof* proteins can interact with various proteins and participate in the activation of gene expression, which makes *Dof* protein show a variety of functions, including tissue development, seed germination, light regulation, and response to biotic and abiotic stresses [4,5].

The first *Dof* gene was cloned from *Zea mays* [6]. To date, numerous *Dof* TFs have been predicted or cloned from higher plants, algae, and bryophytes. The number of *Dof* TFs varied among different species. The highest number of 117 *Dof* genes were identified in *Brassica napus* [7], while only one *Dof* gene was identified in the *Chlamydomonas reinhardt* [8]. It thus can be seen that the *Dof* gene family has expanded dramatically during the evolution from lower plants to higher plants, which may be related to the adaptation to complex environmental conditions. For example, *Arabidopsis Dof5.6* has been shown to be involved in vascular tissue development and interfascicular cambium formation [9].

A lot of studies have shown that the *Dof* TF family plays a variety of functions in promoting growth and development and improving their quality and yield. The overexpression of *ZmDof36* can increase the content of starch and contribute to the nutritional quality of maize [10]. *FaDof2* is regulated by hormones to promote the accumulation of eugenol in strawberries during ripening [11]. The overexpression of *GhDof1* can increase the oil content of upland cotton seeds and reduce the protein content, which is beneficial to the increase in cotton seed oil yield [12]. Recently a study showed that potato CYCLING DOF FACTOR 1 (StCDF1)-*StFLORE* (a long non-coding RNA (lncRNA) counterpart) locus plays an important role in regulating potato vegetative reproduction and water homeostasis [13]. *Dof* TFs also play an important role during seed germination and maturation. Maize PROLAMIN BINDING FACTOR (PBF) and its barley and wheat homologs, BPBF and WPBF, can all participate in the encoding of prolamin [14,15,16]. In addition, protease can hydrolyze storage protein, which contributes to the supply of nutrients to promote seed germination, while the *Dof* gene can induce protease gene expression. For example, BPBF was involved in regulating the expression of cathepsin B-like protease-encoding gene (Al21) [17]. Many *Dof* TFs were involved in plant physiological and biological processes mediated by light signals. Studies have shown that DOF AFFECTING GERMINATION (DAG2) positively regulates the process of light-mediated seed germination [18]. CYCLING DOF FACTOR 1 and 5 (CDF1 and CDF5) were reported to be specifically expressed in short photoperiod to promote hypocotyl elongation [19]. Overexpression of *SlCDF3* caused the late flowering of tomato by inducing the expression of *SlSP5G2* and *SlSP5G3* under the short-day influence and by inducing the transcription of *SlSP5G* under the long-day influence [20]. Previous studies showed that some *Dof* TFs respond to various abiotic stresses such as salt, low temperature, water scarcity, and so on [21]. *SIDof22* induces the expression of ascorbic acid in the tomato and responds to salt stress [22]. The expression of *Brassica napus* Cycling *Dof* Factors 1 (*BnCDF1*) was up-regulated under cold treatment, and the constitutive overexpression of *BnCDF1* enhanced the freezing tolerance of the plant [23]. The overexpression of *BpDof17* enhanced the scavenging ability of reactive oxygen species (ROS), which showed the promotion of drought resistance [24]. Several *Dof* genes were associated with plant responses to biotic stress. *AtDof1.1* (*OBP2*) is a key regulator of glucosinolates biosynthesis in *Arabidopsis*, which plays a defensive role against *Spodoptera littoralis* feeding [25]. Meanwhile, the *Arabidopsis Dof TF OBF BINDING PROTEIN 1* (*OBP1*) genes are involved in the expression of *GST6*, which plays a key function in protecting plant tissues from pathogen damage [26]. Apple *MdDof6* and *MdDof26* genes were reported to be highly induced by the fungus *Alternaria mali* infection [27]. Therefore, a comprehensive analysis of the *Dof* gene family contributes to deeply researching the function and improving the cultivar. Considering the diverse roles of *Dof* TFs in various biological and physiological processes, it is necessary to understand the evolutionary patterns and functional diversity of these TFs in the genomes of important species.

*M. polymorpha* is an annual alfalfa plant, which has the characteristics of good palatability, high nutritional quality, nitrogen fixation, easy sowing, excellent soil improver, and is a nitrogen-fixing agent [28,29]. In the process of growth and development of *M. polymorpha*, lack of water, high salinity, and low temperature are the main limiting factors of its productivity and quality [30], which is not conducive to the large-scale planting of *M. polymorpha* in coastal and cold areas in winter. Recently, our research team was the first to release the genome data of *M. polymorpha*, which provided important research resources for further studies of *M. polymorpha* [31]. However, the function of most genes is not clear, especially the response mechanism of the *Dof* gene to abiotic stress tolerance [31]. Many angiosperm *Dof* genes have been preliminarily identified and functionally characterized [32,33,34,35]. There was a lack of understanding regarding what the similarities or differences are in the structure and function between the *Dof* genes of *M. polymorpha* and other plants. Therefore, we identified the *Dof* gene family members in *M. polymorpha* and analyzed the physical and chemical characteristics, gene structure, conservative motif, phylogenetic relationship, chromosome location, synteny, *cis*-acting elements, protein interaction network, and expression patterns. Our results provide a valuable basis for further understanding the gene expansion and evolution model of the plant *Dof* gene family and analyzing the potential function of the *Dof* gene in plant abiotic stress regulation.

## 2. Results

### 2.1. Identification and Characteristics of the MpDof Gene Family

A total of 36 *Dof* genes of *M. polymorpha* were identified by using 108 *Dof* protein sequences of *O. sativa*, *Arabidopsis*, and *M. truncatula* as a query to perform Blastp, and *MpDof01* was renamed to *MpDof36*, as shown in Table 1. In addition, the physicochemical properties, including the chromosome location, protein length, protein hydrophilic, isoelectric point, protein molecular weight, DNA molecular weight, and orthologous genes of 36 *MpDof* proteins, were detected. The amino acid sequence lengths of these *MpDofs* ranged from 157 to 495 amino acids (Table 1). Correspondingly, the protein molecular weights centered in the range of 17.66–54.15 kDa. The GRAVY values of −0.987 to −0.447 represented that 36 *MpDof* proteins were hydrophilic. Isoelectric point analysis showed that the 7 *MpDof* proteins were acidic (pI < 7), and the remaining 25 *MpDof* proteins were basic proteins (pI > 7).

### 2.2. Comparative Phylogenetic Analysis of Dof Genes in M. polymorpha

To comprehensively understand the number difference of *Dof* genes among four species, phylogenetic relationship analysis and orthologous group classification were performed. The predicted *Dof* proteins were assigned into ten groups named A, B1, B2, C1, C2.1, C2.2, C3, D1, D2, and E as in the *Arabidopsis* classification (Figure 1 and Appendix A). The results also indicated that the number of groups varies greatly among the four tested species. Group B1, B2, C1, C2.1, C2.2, and D1 were found in all four species (Figure 2). Groups A, C3, D2, and E were lineage-specific, while Group E only existed in *O. sativa*. Group B1 and D1 contained more genes than other groups, with 21 and 35 *Dof* genes, respectively. The *Dof* gene numbers in these groups were distinct among different species, for example, two, five, and nine *Dof* genes of group A in *Arabidopsis*, *M. polymorpha*, and *M. truncatula*, respectively. In addition, the phylogenetic analysis based on the genomes of four tested species supported the closest genetic relationship between *M. polymorpha* and *M. truncatula*. The phylogeny also showed that the divergence time between *M. polymorpha* and *M. truncatula* had been estimated to be ~15.3 MYA. The above results suggested that the *Dof* gene family expansions and homologous *Dof* genes gain or lose in *M. polymorpha*, *O. sativa*, *Arabidopsis*, and *M. truncatula* during evolution. Furthermore, the analysis of dicotyledonous model plant *Arabidopsis*, monocotyledonous model plant rice, alfalfa model plant *M. truncatula*, and *M. polymorpha* can provide a basis for studying the evolution of the *Dof* gene in dicotyledonous plants.

### 2.3. Gene Structure and Conserved Motifs Analysis of the MpDofs

To further explore the structural discrepancy among *MpDofs*, exon–intron structures and conserved motifs were analyzed. The phylogenetic relationships of 36 *MpDofs* are shown in Figure 3A, and exon–intron structures are shown in Figure 3B. As the results show, the vast majority of genes in a given subgroup revealed a similar exon–intron structure. Thirty-six *MpDof* genes contained zero or one intron. In subgroup I, nine *MpDof* genes had one intron, and two *MpDof* genes (*MpDof05* and *MpDof31*) had none. Seven *MpDof* genes included in subgroup II had no introns and *MpDof07* had one intron. Among the 17 subgroups III and VI *MpDof* genes, 4 (*MpDof13*, *MpDof28*, *MpDof19*, and *MpDof21*) contained none, and the remaining 13 contained 1.

As shown in Figure 3C, the number of motifs located on each *MpDof* gene varied from 1 to 11. All 36 *MpDof* proteins shared a common motif 1, which corresponds to the *Dof*-conserved domain (Figure 4 and Appendix A). Multiple sequence analysis indicated that 36 *MpDof* proteins contained a highly conserved CX2CX21CX2C single zinc-finger structure, which is significant for the function of *MpDof* genes. A total of 27 of 54 amino acid residues were highly conserved (100% identical in all 36 *MpDof* proteins) amino acids CPRC-S-TKFCY-NNY-QPR-FC-C-R-WT-GG-R-G-R. Moreover, some of the *MpDof* proteins had extra motifs, for example, only *MpDof24* and *MpDof25* contained motif 7 and motif 12, which may be related to specific functions. The motif components of *MpDofs* from subgroup I were the most complex: motif 3, motif 12, motif 19, motif 5, motif 18, motif 14, motif 6, motif 4, motif 17, motif 20, motif 10, motif 7, motif 2, and motif 9 were specific for them. Subgroup III *Dof* proteins possessed relatively simple motif components compared with subgroup I; motif 11 and motif 8 were specific to it. Members belonging to subgroups II and IV contained only motif 1, except for *MpDof02* and *MpDof20*, which both contained motif1 and motif 16. Thus, the results suggested that differences in exon–intron distribution and motif components between subgroups may lead to functional diversity.

### 2.4. The Location on Chromosome, Gene Duplication, and Syntenic Analysis

The location of *MpDof* genes on chromosomes and synteny in the *M. polymorpha* genome are shown in Figure 5 and Appendix A. The distribution of *MpDof* family members on 7 chromosomes was uneven and each chromosome possessed *MpDof* genes ranging from 2 to 7 (Figure 5). Chromosomes 4 and 7 had the fewest number of *MpDof* genes with two, while chromosomes 1, 3, and 6 had the largest number of *Dof* genes with 7. We found a total of 15 pairs of *MpDof* gene duplications which included 13 segmental and 2 tandem duplications (Figure 5). We can see that some *MpDof* genes, such as *MpDof11*, have been duplicated three times. Tandem duplications led to *MpDof* gene clusters or hot regions, such as the *MpDof* gene clusters located on chromosomes 5 and 7, whereas segmental duplications caused homologous genes between chromosomes that may expand the number of *MpDof* genes groups. Moreover, the synteny relationship between *M. polymorpha* and *M. truncatula* was stronger (Figure 6), while the synteny relationship with rice was the simplest. This can provide information on the origin and evolution of species.

The non-synonymous substitution rate (Ka), synonymous substitution rate (Ks), and Ka/Ks value of 15 duplicated gene pairs were calculated to report the natural selection pressure acting on *Dof* gene pairs and to estimate the date of *MpDof* gene duplication. We found that the Ka/Ks ratios of all 15 *MpDof* duplicated gene pairs were less than one (Table 2). This suggested that these duplicated gene pairs were under negative selection pressure. The Ka values were used to calculate the date on which 15 *MpDof* gene pairs duplicated. The results showed that the duplicated events of 15 *MpDof* gene pairs spanned approximately from 1.79 to 93.36 MYA (millions of years ago) (Table 2). In addition, we can also observe that the Ka and Ks values of tandem-duplicated gene pairs (*MpDof24*/*MpDof25*; *MpDof35*/*MpDof36*) were the lowest, and the divergent time of them was later than 13 segmental-duplicated gene pairs.

### 2.5. Cis-Acting Elements Analysis

We mainly analyzed light-responsive, hormone-responsive, abiotic-stress-responsive, and development-related elements (Appendix A), which are the key regulatory factors involved in plant growth and development or other important physiological processes. The *cis*-acting elements were distributed among these predicted *MpDof* genes. Analysis of the *cis*-acting elements can be conducted to estimate the putative functions of *MpDof* genes. Compared with *cis*-acting elements related to abiotic stress and development, there were more hormone-responsive elements in the promoter regions of 36 *MpDof* genes, including those related to auxin, gibberellin, salicylic acid, abscisic acid, and methyl jasmonic acid, while abscisic acid- and methyl jasmonic acid-responsive elements were the most abundant (Figure 7 and Appendix A). It is suggested that the *Dof* genes may regulate plant growth and development or other physiological processes mainly by responding to plant hormone signals, especially abscisic acid and methyl jasmonic signals. We predicted that *cis*-acting elements related to abiotic stress, including DRE, LTR, WUN-motif, GC-motif, MBSI, TC-rich repeats, and MBS, and elements involved in defense and stress-responsiveness and drought inducibility (TC-rich repeats and MBS) were relatively abundant. Among the development-related elements identified, the number of elements involved in endosperm expression, meristem expression, and meristem expression were dominant. In addition, it was also found that there are few light-responsive elements in the *Dof* gene family of *M. polymorpha*.

### 2.6. MpDofs Protein–Protein Interaction Network Analysis

The STRING online tool was employed to predict the functional relationships of *MpDof* proteins by analyzing protein–protein interactions among the *Arabidopsis* homologs of *MpDofs*. The amino acid sequence of DOF5.5 was highly similar to those of *MpDof24* and *MpDof25*; the sequence of CDF2 was highly similar to those of *MpDof05*, *MpDof14*, *MpDof17*, *MpDof27*, *MpDof35*; the sequence of CDF3 was highly similar to those of *MpDof11*, *MpDof26*, *MpDof36*; and the sequence of AT5G02460 was highly similar to those of *MpDof10* and *MpDof16* (Appendix A). The DOF5.5, CDF2, CDF3, and AT5G02460 all had complex interactions with FKF1 (Figure 8). In addition, there were interactions between *MpDof24* and *MpDof25* and *MpDof11*, *MpDof26* and *MpDof36*. Similarly, OBP1 and AT2G34140, which were highly similar to *MpDof03* and *MpDof31*, respectively, were predicted to exist in interactions with TGA4. Moreover, OBP3 (*MpDof08*, *MpDof13*, *MpDof15*, *MpDof28*, *MpDof34*) interacted with TGA4 and FKF1.

### 2.7. Expression Patterns of MpDof Genes at Different Development Stages, Tissues, and Abiotic Stresses in Medicago Plants

The *MpDof* gene expression data for different developmental stages (seeding stage; early flowering stage; late flowering stage) were visualized and analyzed. As shown in Figure 9A and Appendix A, there were divergences among 36 *MpDof* genes for expression levels in different developmental stages. *MpDof26* showed the maximum expression level at the early flowering stage, while the minimum expression level was shown at the seeding stage. *MpDof01* had a comparatively high expression level in all three growth stages. Some *MpDof* genes, such as *MpDof08*, *MpDof17*, and *MpDof25*, were not expressed in the three growth stages. We predicted the expression pattern of *MpDof* genes in different developmental tissues (leaf bud, large pod, leaf, medium pod, flower, petiole, root, small pod, stem). The expression patterns suggested that *MpDof* genes have distinct transcript levels in different tissues. The 36 *MpDof* genes were classified into 4 subgroups based on their expression patterns. As shown in Figure 9B and Appendix A, subgroup A contained seven *MpDof* genes that were expressed in any tissue. In subgroup B, *MpDof29* was specifically expressed in large pods and *MpDof31* showed specific expression in the leaf, while the remaining five *MpDof* genes were not expressed in all nine tissues. Subgroup C contained 11 *MpDof* genes, which were mainly expressed in the flower, root, and stem, but the expression level was relatively low. Subgroup D showed that *MpDof27* and *MpDof28* were expressed in the leaf and root, and that *MpDof32* was highly expressed in the flower and large pod.

We assessed the expression patterns of the *M. sativa* homologous *Dof* genes in leaves and roots under salt and drought stress. Under drought stress (Figure 9C and Appendix A), *MpDof01* was highly expressed in both leaf and root tissues. It is worth noting that most *MpDof* genes had significantly low expression levels in leaves after drought treatment. However, some *MpDofs* were significantly up-regulated after drought stress in roots, such as *MpDof02*, *MpDof03*, *MpDof15*, *MpDof20*, and *MpDof27*. *MpDof11*, *MpDof26*, and *MpDof27* were obviously down-regulated in both tissues under salt stress (Figure 9D and Appendix A). In addition, the expression of *MpDof02*, *MpDof03*, *MpDof09*, *MpDof10*, and *MpDof34* in the root was clearly higher than in the leaf. However, the peak expression of these genes occurred at different time points after salt treatment: the expression peaks of *MpDof02* and *MpDof09* appeared at 27 h, while the expression peaks of *MpDof03*, *MpDof10*, and *MpDof34* appeared at 3 h.

### 2.8. Validation of MpDof Genes Expression Patterns under Abiotic Stress

All eight *MpDof* genes had different responses to drought, salt, and low-temperature stress. The changes in five genes (*MpDof10*, *MpDof22*, *MpDof23*, *MpDof26*, and *MpDof29*) in drought response were essentially consistent, they were obviously up-regulated at 12 h, but their expression levels were not significantly different from those of the control (0 h) at 24 h. In addition, the expression of *MpDof03* gradually increased under drought conditions (Figure 10). Compared with the control, the expression level of *MpDof03* was up-regulated by 3 times at 24 h. There was no significant difference in the expression of *MpDof11* at other drought treatment time points compared to the control. *MpDof25* was highly induced at 3 h, and the relative expression level was about 140 times that of the control. During salt treatment, the expression levels of eight *MpDof* genes at 12 h and 24 h were significantly higher than those at 0 h and 3 h. *MpDof11* was significantly down-regulated after the low-temperature treatment. *MpDof11* was down-regulated about 5 times compared with the control after 24 h of low-temperature treatment. *MpDof03*, *MpDof22*, and *MpDof23* were not induced by cold stress. The expression peaks of *MpDof10* and *MpDof25* both appeared at 3 h, which were about 2.5 times and 4 times that of the control, respectively. Furthermore, the peaks of *MpDof26* and *MpDof29* occurred at 12 h, and then gradually decreased, which were about 36 times and 18 times that of the control, respectively.

## 3. Discussion

The *Dof* TFs have been shown to play transcriptional regulatory roles in numerous important physiological and biochemical processes in plants, such as nutrient transport, carbon and nitrogen metabolism, and plant hormones [36,37,38]. As a multifunctional vegetable, *M. polymorpha* has an important application value [29]. However, due to its limited distribution, some adverse environmental conditions in many areas are not conducive to its normal growth and development. To provide new gene resources for the cultivation of new resistant varieties of *M. polymorpha*, we need to conduct a comprehensive investigation of the *Dof* gene family. In this study, we discussed the possible evolutionary patterns of the *Dof* family of *M. polymorpha* and determined their potential functions under abiotic stress. Thirty-six *Dof* genes were identified in *M. polymorpha*. Forty-two, thirty-six, and thirty *Dof* genes were identified in *M. truncatula* [39], *Arabidopsis* [40], and *O. sativa* [33], respectively. However, the genome sizes of *M. polymorpha*, *M. truncatula*, *Arabidopsis*, and rice are about 457 Mb, 500 Mb, 125 Mb, and 466 Mb, respectively, which means that the number of *Dof* TF family members does not increase with the size of the genome. The above results also suggested that *Dof* genes have shown greater diversity in long-term evolution.

The number of *Dof* genes of *M. polymorpha* and many angiosperms such as *M. truncatula*, *M. sativa*, *Arabidopsis*, and rice are close to each other [39,41,42]. However, previous studies found that a single *Dof* gene was found in *Chlamydomonas Rheinis*, 19 *Dof* genes were found in moss *Physcomitrella patens*, and *Dof* genes were not identified in the red algae *Cyanidioschyzon merolae* and the diatom *Thalassiosira pseudonana* [43], suggesting that *Dof* genes may originate from the earlier ancestors before the differentiation between *Chlamydomonas* and terrestrial plants. In addition, 12 and 10 *Dof* genes were identified in fern *Selaginella moellendorffiii* and gymnosperms *Pinus taeda*, respectively [43]. Therefore, we speculate that ferns and gymnosperms have *Dof* gene loss events after the differentiation of moss lineage and vascular plants, while large-scale gene duplication events of *Dof* genes occur during the evolution of angiosperms, which may be closely related to the development of complex regulatory mechanism networks of angiosperms.

The phylogenetic relationship analysis is conducive to our preliminary understanding of the evolution of gene families and gene functions [44]. Similar to previous studies [40,41,45], the 36 *Dof* genes in *M. polymorpha* were mainly divided into 10 groups. *MpDof* gene numbers of the orthologous groups are different among four species, indicating that the *Dof* gene family may experience various evolutionary pathways in distinct species. Group D1 contained the largest number of *Dof* genes, with 35, while groups C2.2 and D2 contained only 8 and 4 *Dof* genes, respectively, suggesting that these orthologous groups had different degrees of gene amplification and loss. Phylogenetic analysis showed that *MpDof* genes had the closest genetic relationship with *MtDof* genes, but were further related to *Arabidopsis* and *O. sativa*. These *MpDof* genes may be produced by further genomic expansion after the divergence of different plant species, thus legume plants are more closely related. Group E only included 12 *OsDof* genes, indicating that group E evolved separately during the differentiation of monocotyledons and dicotyledons. The above results also support the theory that genes continue to experience random loss or obtain events in the process of evolution [31,46]. The *Dof* members in the same group may have similar functionality [4]. *Arabidopsis Dof* members (AtDof9, AtDof15, AtDof30, and AtDof32) included in group C1 have been shown to be involved in plant vascular tissue and seed development and are highly expressed in root tissues [9,47,48]. It was also found that *MpDof09*, *MpDof18*, and *MpDof33* included in group C1 were involved in extensive tissue expression. We speculate that the function of *MpDofs* is similar to that of *Arabidopsis* CDF protein included in D1 group, which is a key factor in regulating photoperiod flowering in *Arabidopsis* and plays an important role in different abiotic stress responses [49,50,51,52]. Furthermore, protein network interaction predicted that the *MpDof* protein in group D1 may play a role in plant photoperiod flowering and abiotic stress response through interaction.

Further duplication analysis showed that segmental duplication played a major role in the expansion of *Dof* gene in *M. polymorpha*. This result is similar to the *Dof* family reported in *M. truncatula* and Tartary buckwheat [39,53]. However, some gene families (CAB family) had high-level tandem duplications, which functionally differ from these (MYB TF) with high segmental duplication levels [54]. In addition, divergence times of all the segmental duplication gene pairs ranged from 21.45 to 93.36 MYA. It can be concluded that all the segmental-duplicated gene pairs may occur after the differentiation of *Arabidopsis* and legumes and before the differentiation of *M. polymorpha* and *M. truncatula*. However, tandem-duplicated gene pairs occur after the formation of *M. polymorpha* and may play a special role in *M. polymorpha*. Furthermore, the *MpDof* genes have undergone negative selection during the evolution of *M. polymorpha*. The diversity of the Ka/Ks value indicates that the evolution rate of duplicated genes is different when facing the pressure of purification selection, which may lead to the functional differentiation of the *Dof* gene [55]. There are great differences in tissue expression between *MpDof14* and *MpDof25*, suggesting that some duplicated genes had functional differentiation. The expression patterns of *MpDof11*, *MpDof14*, and *MpDof27* in different developmental stages of *M. polymorpha* are similar, indicating that some genes may have functional redundancy after duplication.

The genes containing the same motif are most likely caused by gene duplication events [5]. Furthermore, the result that the *Dof* gene of the same subgroup has a similar exon–intron structure can further support the close evolutionary relationship among the *Dof* members in the evolutionary tree. The *Dof* gene of *M. polymorpha* contains very few introns (zero to one intron), which is similar to *Arabidopsis*, tea, and cassava *Dof* genes [40,45,56]. However, *Chlamydomonas*, *Physcomitrella patens*, *Selaginella moellendorffiii*, and *Pinus taeda* contain 4, 6, 5, and 4 introns, respectively [8]. In *Medicago* plants, the number of introns in *M. truncatula* and *M. sativa Dofs* varies from zero to four, and zero to seven, respectively [39,41]. The results revealed that intron loss occurred during the evolution of *M. polymorpha*. Studies have shown that the evolution of poor introns and intron-free genes may be related to vascular system and seed development, which promotes the complexity of structure and regulatory pathways during the evolution of higher plants [57]. In addition, previous studies of gene families have found that genes with no intron and intron deficiency (three or less) are more likely to play a role in abiotic stress responses such as drought and salt than intron-rich genes [58]. However, further experiments are needed to analyze the specific functions of poor intron family genes in plant growth and development and resistance to abiotic stress. The *MpDof* gene family can be used as a resource for a poor-intron gene family, which provides important value for exploring the origin, evolution, and function of plants.

In recent decades, numerous studies have reported that the plant *Dof* TFs are involved in the plant’s response to various abiotic stresses. The results revealed that the overexpression of *SlCDF1* or *SlCDF3* enhanced *Arabidopsis* plant resistance to drought and salt, and stress-responsive genes such as *COR15*, *RD29A*, and *RD10* were transcriptionally activated, suggesting that *SlCDF* may be a significant upstream regulator of the plant’s response to drought and salt stress [59]. The *OsDof1* overexpression line ensured the seed-setting rate under low-temperature stress, demonstrating that it participated in the biological process of enhancing cold tolerance in rice [60]. *MpDof01* was expressed by drought in leaves and roots, indicating that *MpDof01* may help to improve drought tolerance in plants. Furthermore, some *MpDof* genes may have diverse regulatory mechanisms in different tissues and abiotic stress responses. For example, the expression of *MpDof03* is up-regulated in roots and down-regulated in leaves under salt stress, while *MpDof03* is down-regulated in roots after salt stress. qRT-PCR results further showed that the expression levels of most *MpDofs* changed significantly under drought, salt, and cold stress, indicating that these genes play a key role in improving abiotic stress resistance. The validated eight *MpDofs* were differentially expressed under three types of abiotic stress. For example, *MpDof11* had no significant change under drought treatment, was significantly up-regulated under salt stress, and was obviously down-regulated under low-temperature conditions, suggesting that the regulation mechanism of *MpDof* genes were different under different stresses. In *Brassica napus*, the expression level of *BnCDF1* was not affected by drought or low-temperature stress, while the *BnCDF1* transgenic *Arabidopsis* showed higher freezing resistance [23]. Salt treatment significantly increased the expression of all eight *MpDof* genes, showing that *MpDofs* may be important regulatory factors for *M. polymorpha* in response to salt stress. *Arabidopsis* that overexpressing *MtDof32*, which was divided into group D1, had stronger salt tolerance than the wild-type plants [61]. In addition, soybean *GmDof41* positively regulated salt tolerance by binding with the promoter of *GmDREB2A* [62]. It is worth noting that *MpDof25* was induced by drought, salt, and low temperature, but its expression increased by about 140 times in 3 h of drought treatment, which suggested that *MpDof25* may be a drought-sensitive gene. Similarly, *MpDof26* was considerably up-regulated under three stress treatments, indicating that it may be a crucial TF for plants to respond to diverse abiotic stresses and can be used as a potential target for plant stress resistance breeding. Overexpression of AT3G47500.1 (*AtCDF3*), which is highly homologous to *MpDof26*, improves the tolerance of *Arabidopsis* to drought, salt, and cold stress [51]. These results show that *MpDof* TFs are induced by multiple abiotic stresses; thus, we speculate that *MpDofs* may have complex regulatory functions in abiotic stresses.

To sum up, this study contributes by further improving the evolutionary path of the *Dof* gene family in dicotyledons and provides a basis for the study of the structural and functional diversity of *Dof* genes. In addition, our results found that *MpDof10*, *MpDof25*, *MpDof26*, and *MpDof29* may be good candidates for genetic engineering breeding with regard to resisting various abiotic stresses.

## 4. Materials and Methods

### 4.1. Genome-Wide Identification of Dof Transcription Factor in M. polymorpha

The genome, CDS, transcript, and protein sequences of *M. polymorpha* were downloaded from the National Genomics Data Center (https://ngdc.cncb.ac.cn/search/?dbId=gwh&q=Medicago_polymorpha, accessed on 2 September 2021). The *Dof* protein sequences of *O. sativa*, *Arabidopsis*, and *M. truncatula* were obtained from the Plant Transcription Factor Database v5.0 (http://planttfdb.cbi.pku.edu.cn/, accessed on 2 September 2021), and these sequences were used as queries to search against *M. polymorpha* protein and the CDS database using BLASTp, with expected values of 1e^−5^ [63]. The *MpDof* protein containing the conserved *Dof* domain was identified based on the HMMER-HELP from the Pfam (http://pfam.xfam.org, accessed on 4 September 2021) website with an E-value of 1.0. Then, non-redundant sequences were obtained by utilizing EXPASY (https://web.expasy.org/protparam/, accessed on 4 September 2021). Finally, the molecular weight (MW), theoretical isoelectric point (pI), amino acid length, grand average of hydropathicity (GRAVY) of the *Dof* protein, and molecular weight of DNA were evaluated by using the SMS2 online tool (http://www.detaibio.com/sms2/index.html, accessed on 4 September 2021). Furthermore, the orthologs of *MpDof* proteins in *Arabidopsis* were predicted using the PlantTFDB online website (http://planttfdb.gao-lab.org/prediction.php, accessed on 7 October 2021).

### 4.2. Phylogenetic Analysis of Dof Family Members and Orthologous Groups Identification

To comprehensively acquaint the evolutionary relationships of the *MpDof* gene family, a multiple-sequence alignment of 141 *Dof* protein sequences from *Arabidopsis*, *O. sativa*, *M. truncatula*, and the putative *MpDof* protein sequences was conducted using ClustalW with the default parameters. Subsequently, the phylogenetic tree was constructed by the neighbor joining method with 1000 bootstrap replicates in MEGA7 software (with a Poisson model and Pairwise deletion).

In this study, diamond software was used to perform an all-versus-all BlastP search with parameters: E value 1 × 10^−3^ as the input file for the OrthoFinder software [64], and orthologous groups were identified by previously published methods [65]. Then, Tajima’s D values of all orthologous groups were calculated by DnaSP 5.0 to evaluate the selective forces among orthologous groups [66].

### 4.3. Detection of Gene Structures and Conserved Motifs of MpDofs

Both CDS sequences and genomic sequences of *MpDof* were used to predict the *MpDof* gene structure by utilizing the Gene Structure Display Server (http://gsds.gao-lab.org/, accessed on 14 November 2021). The intron and exon distribution were exhibited in the gene structure. The highly conserved *Dof* domain was identified based on multiple sequence alignment using DNAMAN version 7 software. The *MpDof* protein sequences were identified using the MEME online program v5.4.1 (https://meme-suite.org/meme/, accessed on 14 November 2021), and the maximum number of motifs was 20. Ultimately, the gene structure and motifs were visualized using TBtools software [67].

### 4.4. Chromosomal Localization, Gene Duplication Events, and Synteny Analysis of the MpDof Genes

The chromosomal position of the *MpDof* gene was obtained from the gff3 file in the Genome Warehouse in National Genomics Data Center (https://ngdc.cncb.ac.cn/search/?dbId=gwh&q=Medicago_polymorpha, accessed on 2 September 2021). The “MCScanX” function was used to analyze gene duplication events. Then, the chromosome location of 36 *Dof* genes and synteny relationships in the *M. polymorpha* genome were visualized with “Advanced Circos” tools. Syntenic analysis images of *M. polymorpha* between *Arabidopsis*, *M. truncatula*, and *O. sativa* were conducted by “Dual Systeny Plot” view software in TBtools.

The nonsynonymous (Ka), synonymous (Ks), and Ka/Ks ratios for 15 duplicated *MpDof* gene pairs were calculated by performing the Simple Ka/Ks Calculator function of TBtools software. It is generally assumed that Ka/Ks > 1, Ka/Ks = 1, and Ka/Ks < 1 indicate positive selection pressure, natural selection pressure, and negative selection pressure, respectively [68]. Then, the time of duplication (T) was estimated using the formula T = Ks/2λ, where λ = 1.5 × 10^−8^ represents the rate of replacement of each locus per dicotyledon year [69].

### 4.5. Identification of Promoter Cis-Acting Elements

The promoter of 2000 bp upstream sequences from the initiation codon of the putative *MpDof* genes was extracted. Then, the *cis*-acting elements in these sequences were predicted in the PlantCARE database (http://bioinformatics.psb.ugent.be/webtools/plantcare/html/, accessed on 14 October 2021) and visualized utilizing TBtools software [70].

### 4.6. Functional Protein Interaction Network Prediction of MpDofs

In the study, *Arabidopsis* was used as the association background to analyze the initial function of *MpDofs*. The putative *MpDof* protein sequences were located in the STRING server (https://cn.string-db.org/ accessed on 12 November 2021) to predict the functional interaction networks and functional annotations. The minimum required interaction score was 0.70, and the first and second shells were set to no more than 10 interactors. The network edges were set as line colors to represent the protein–protein interaction types.

### 4.7. Expression Analysis of Dof Genes among Different Tissues at Different Development Stages

The expression pattern of each *MpDof* gene in the seedling stage, early flowering stage, and late flowering stage were obtained from the papers published by our research team [31].

As a model plant in legumes, *M. truncatula* has a high sequence similarity with *M. polymorpha*. The gene expression data of *M. truncatula* in different organs including leaf bud, large pod, leaf, medium pod, flower, petiole, root, small pod, and stem were downloaded from LegumeIP v3 Browser (https://www.zhaolab.org/LegumeIP/gdp/13/gene/profile/5?sessionid=Plant_organs, accessed on 5 December 2021) to predict the organ-specific expression patterns of *MpDof* genes at diverse development stages [71]. The expression profile was shown in the heatmap function in TBtools.

### 4.8. Expression Pattern Analysis of MpDof Genes under Abiotic Stress

Because *Medicago sativa* is a perennial *Medicago* plant, Alfalfa Gene Editing Database (http://alfalfagedb.liu-lab.com/heatmap/heatmap/, accessed on 12 February 2023) was used to retrieve *M. sativa Dof* genes homologous to *MpDof* genes and to evaluate the expression profiles of these Ms*Dof* genes under abiotic stress. The heatmap drawing tool is the same as above.

To further explore the expression level of *MpDof* genes under different abiotic stress treatments, eight *MpDof* genes were randomly selected according to the orthologous genes responding to abiotic stress for qPCR verification. *M. polymorpha* seedlings were collected after drought (PEG 6000, 20%), salt (220 mM), and low-temperature (4 °C) treatments for 0, 3, 12, and 24 h, respectively, and three replicates of each sample were applied to perform qRT-PCR. The total RNA was extracted using a Vazyme Total RNA Isolation Kit (Vazyme, Nanjing, China). The total RNA was synthesized into the first strand of cDNA using the cDNA synthesis kit (Vazyme, Nanjing, China). Eight pairs of gene-specific primers for qRT-PCR analysis were designed by PerlPrimer v1.1.21 software and displayed in Appendix A. A 10 μL reaction volume for each sample contained 5 μL of 2 × AceQ Universal SYBR qPCR Master Mix (Vazyme, Nanjing, China), 0.2 μL of each primer, 1 μL of diluted cDNA product, and 3.6 μL of ddH_2_O. The qRT-PCR reaction procedures recommended in the qPCR kit, which were carried out on the platform supported by the QuantStudio 3 system (Thermo Fisher Scientific, Waltham, MA USA), were as follows: 5 min at 95 °C for DNA polymerase activation, denaturation, and anneal/extension at 95 °C for 10 s and 60 °C for 30 s, respectively, for a total of 40 cycles. *Mt-ubiquitin* was selected as the internal control to calculate the relative expression data according to the 2^−∆∆*CT*^ method.

## 5. Conclusions

In this study, we identified 36 *Dof* genes in *M. polymorpha*, and the results showed that these *Dof* genes all have highly conserved single zinc finger domains. At the same time, phylogenetic relationships, conserved motifs, the exon–intron structure, synteny, gene duplication events, *cis*-acting elements, protein interaction, and expression profiles were comprehensively analyzed. The protein–protein interaction network analysis indicated that multiple *MpDof* proteins may participate in regulating photoperiod flowering. The expression profiling analysis showed that *MpDof* genes possessed different expression patterns in distinct tissues and developmental stages. Combined with *cis*-acting element analysis, the possible regulatory expressions of *MpDof* genes were further detected. In addition, the expression of most *MpDofs* presents different dynamic reaction processes under drought, salt, and low-temperature conditions. The selected eight *MpDof* genes were significantly up-regulated under salt treatment, which proved that the *MpDof* genes played an important role for *M. polymorpha* in responding to salt stress. Our results would provide references for the further study of various functions and regulatory networks of *Dof* genes in *M. polymorpha*.

## Figures and Tables

**Figure 1 plants-12-01831-f001:**
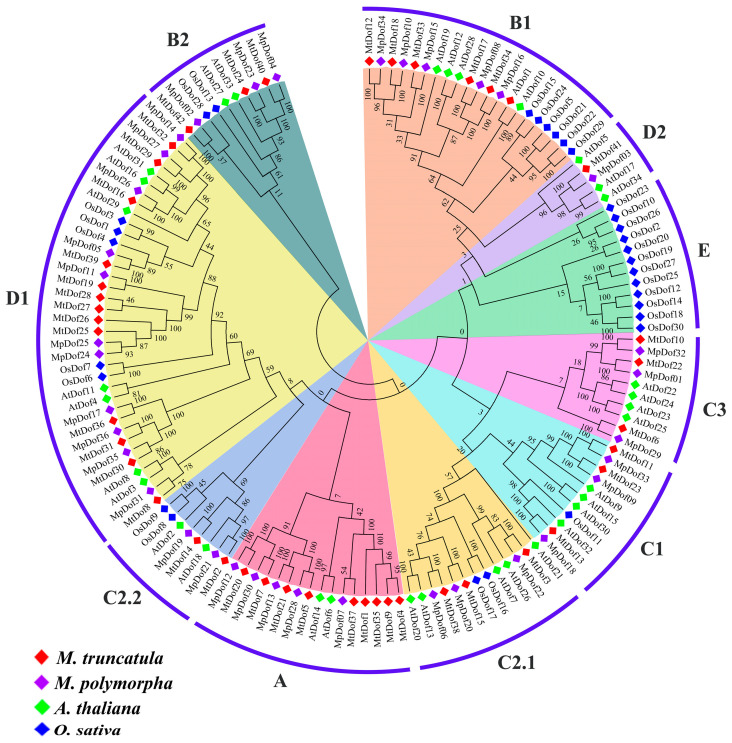
Phylogenetic relationships of the Dof proteins from *M. polymorpha*, *M. truncatula*, *A. thaliana*, and *O. sativa*. The *Dof* genes of *M. truncatula*, *M. polymorpha*, *A. thaliana*, and *O. sativa* were marked with red, purple, green, and dark blue diamonds, respectively.

**Figure 2 plants-12-01831-f002:**
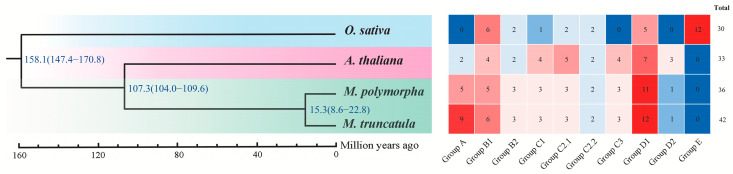
The phylogenetic tree and *Dof* genes distribution among *O. sativa*, *A. thaliana*, *M. polymorpha*, and *M. truncatula*. The numeric value beside each node shows the estimated divergence time of different species. The number of *Dof* genes in all groups are presented by a heatmap.

**Figure 3 plants-12-01831-f003:**
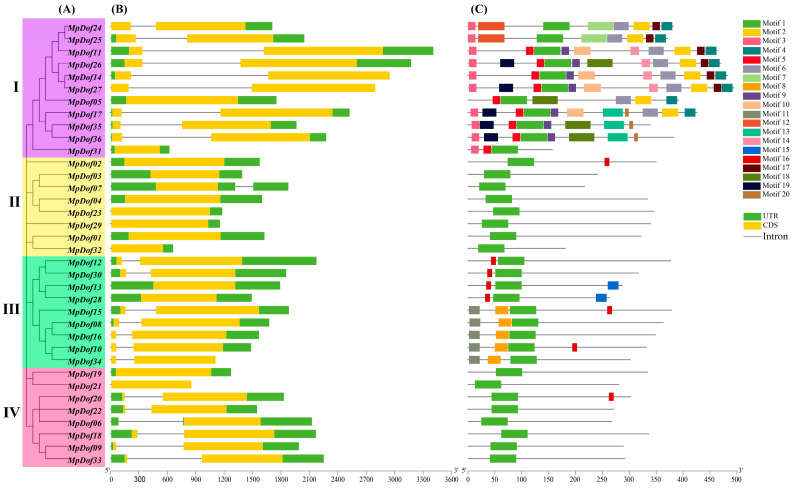
The phylogenetic relationships (**A**), gene structure (**B**), and motif composition (**C**) of *MpDof* proteins.

**Figure 4 plants-12-01831-f004:**
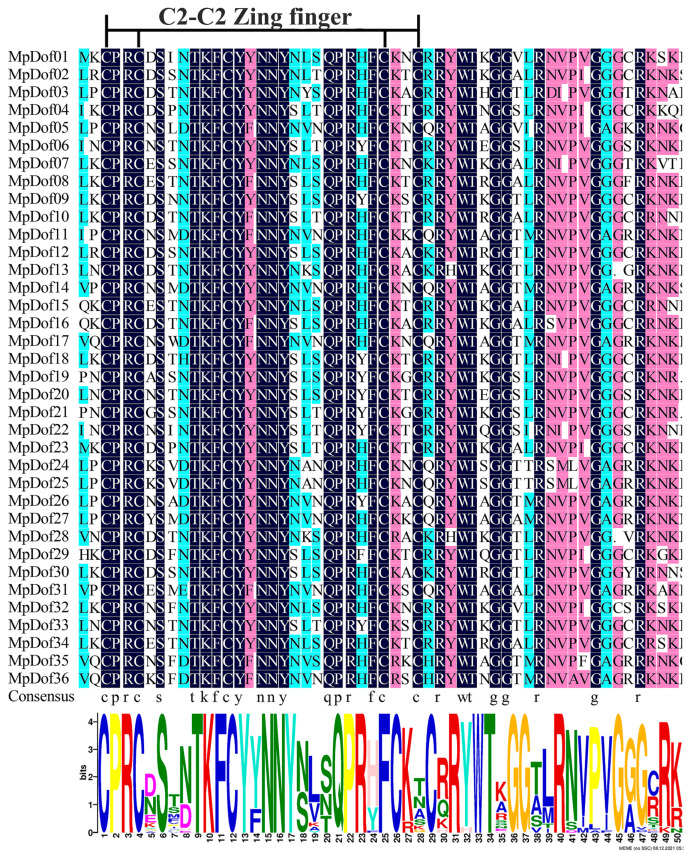
Multiple sequence alignment of the DNA-binding domain in MpDof proteins. The identical amino acids are shown at the bottom and the four cysteine residues are displayed on top. A LOGO diagram represents the conservativeness of the amino acid sequence.

**Figure 5 plants-12-01831-f005:**
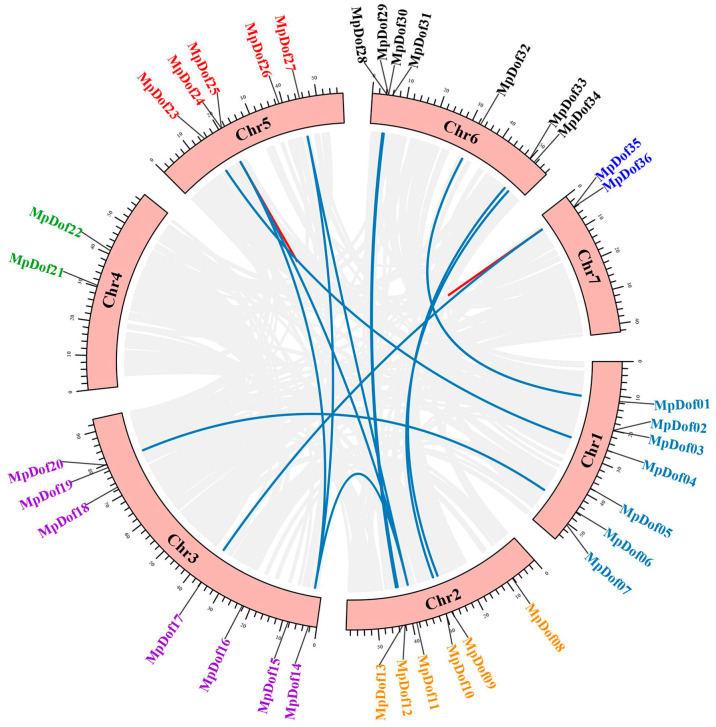
Chromosomal location and gene duplication events of *MpDof* genes. Segmentally and tandemly duplicated *MpDof* gene pairs are indicated with midnight-blue lines and red lines, respectively.

**Figure 6 plants-12-01831-f006:**
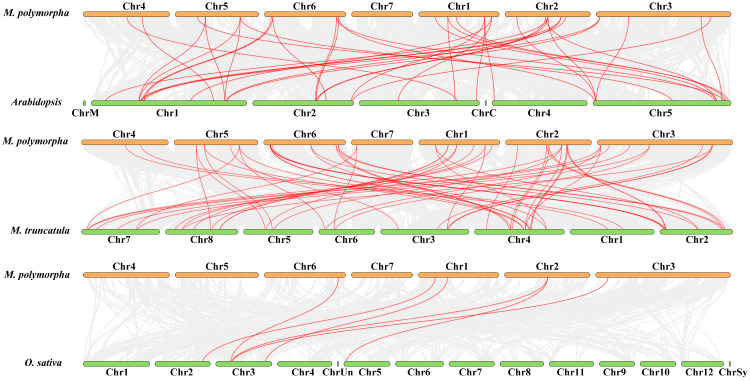
The synteny analysis of the *Dof* genes between *M. polymorpha* and *A. thaliana*, *M. truncatula*, and *O. sativa*, respectively. The gray line represents collinear blocks, and the red line shows the orthologous gene pairs between *M. polymorpha* and *Arabidopsis*, *M. truncatula*, and *O. sativa*, respectively.

**Figure 7 plants-12-01831-f007:**
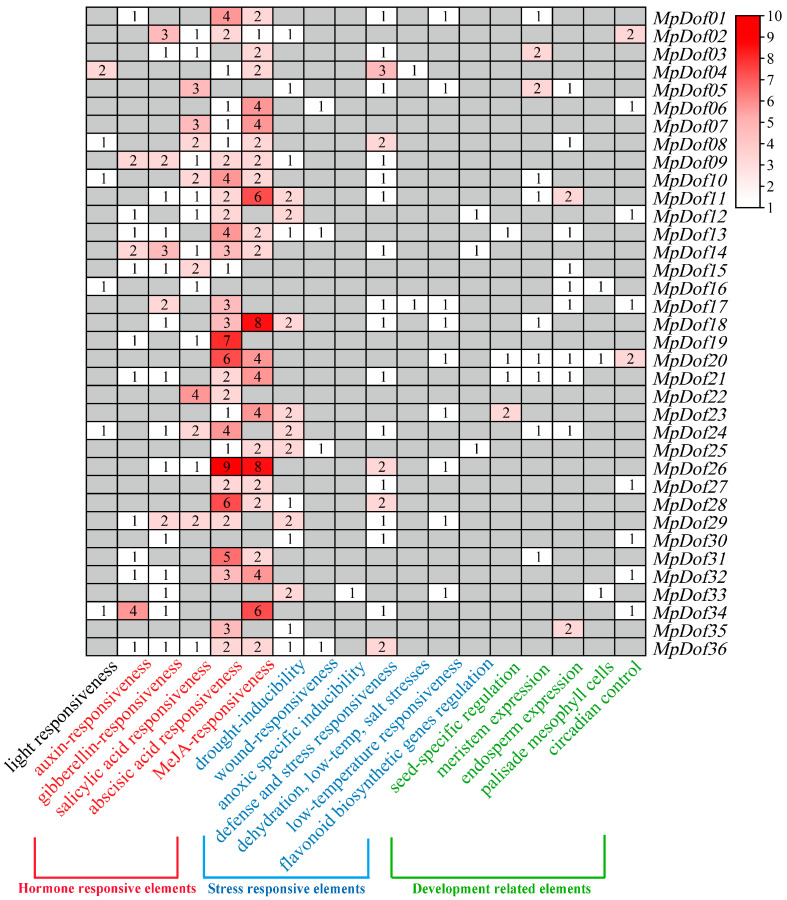
Distribution of *cis*-acting elements from the *MpDof* genes in *M. polymorpha*. The numbers represent the sum of diverse *cis*-acting elements contained in *MpDof* genes.

**Figure 8 plants-12-01831-f008:**
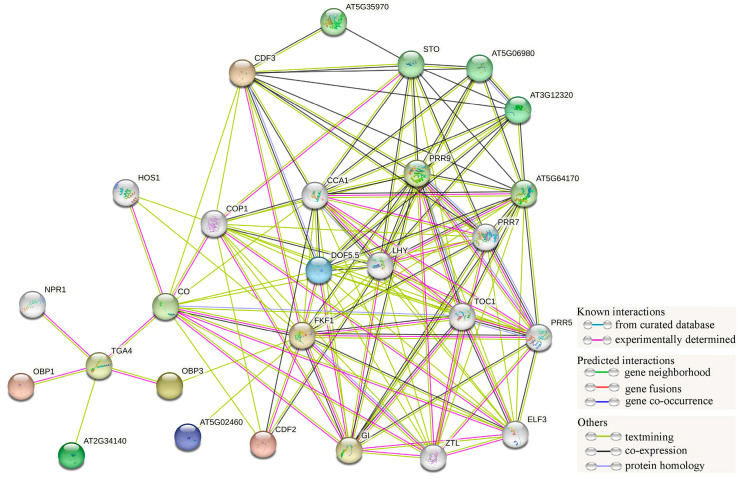
Protein–protein interaction network of *Dofs* in *M. polymorpha* according to the orthologs in *Arabidopsis*. Different colored lines represent the type of interactions, which are displayed in the legend.

**Figure 9 plants-12-01831-f009:**
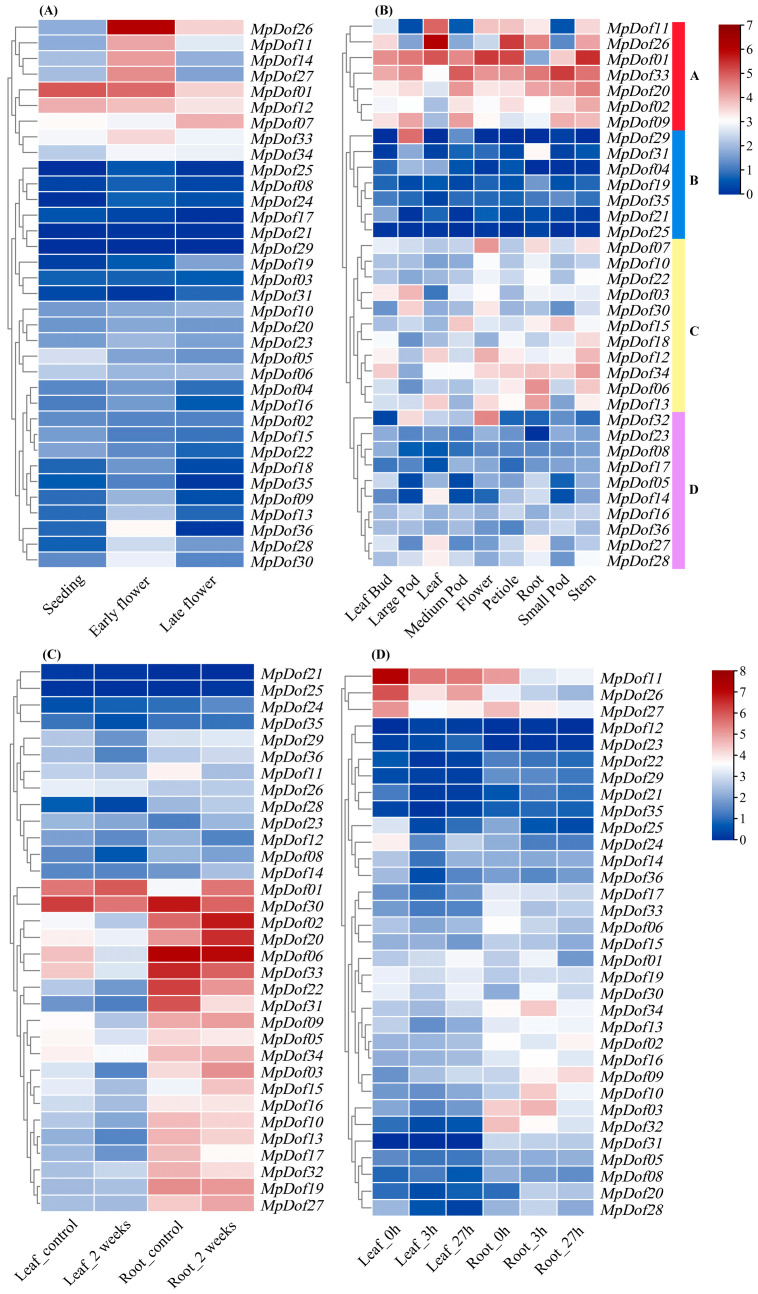
Heatmap display of the *Dof* genes in *Medicago* plants. (**A**) The expression levels of *M. polymorpha Dof* genes in leaves at three growth stages; (**B**) the expression profiles of *M. truncatula* homologs of *MpDof* genes in different developmental tissues. (**C**) The expression profiles of *M. sativa* homologous *Dof* genes in leaves and roots after two weeks of water deficit treatment. (**D**) The changes in each gene expression in leaves and roots at different time periods after salt exposure. The color scale indicates change folds from higher (red color) to lower (blue color).

**Figure 10 plants-12-01831-f010:**
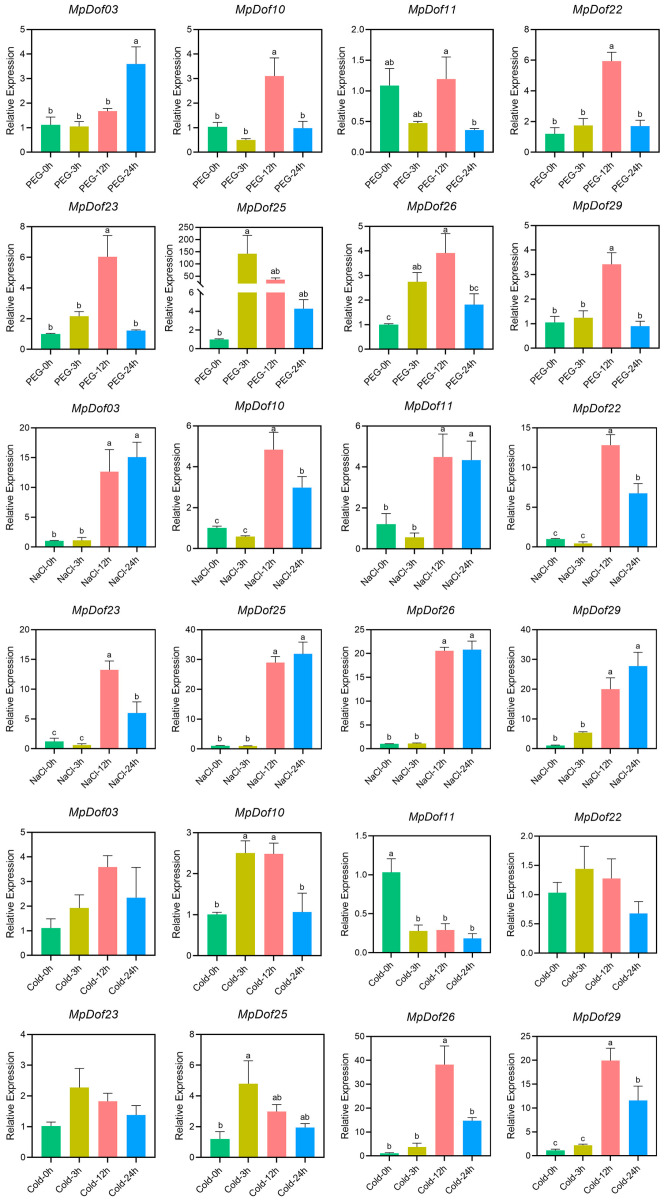
The relative expression levels of *MpDof* genes in leaves under drought, salt, and low-temperature stress. Relative expression data were calculated using *Mt-ubiquitin* as the reference. Different lowercase letters represent significant differences between different treatment time points (*p* < 0.05).

**Table 1 plants-12-01831-t001:** The asic information of *MpDof* genes family identified in *Medicago Polymorpha*.

Gene Name	Gene ID	Chromosome Location	Protein Length (aa)	Protein GRAVY ^1^	Isoelectric Point (pI)	Protein Molecular Weight (kDa)	DNA Molecular Weight (Da)	Orthologous
*MpDof01*	Mpo1G10010	Chr1.1: 11559125-11560093: +	322	−0.804	6.73	34.64	299,476.22	AT5G60850.1
*MpDof02*	Mpo1G16410	Chr1.1: 20059424-20060476: −	350	−0.651	8.48	37.93	324,040.69	AT5G65590.1
*MpDof03*	Mpo1G16460	Chr1.1: 20147292-20148017: +	241	−0.447	8.49	25.28	224,442.30	AT3G50410.1
*MpDof04*	Mpo1G20440	Chr1.1: 25979040-25980044: −	334	−0.638	8.26	36.64	310,622.49	AT5G65590.1
*MpDof05*	Mpo1G25080	Chr1.1: 37942310-37943488: −	392	−0.648	8.05	43.28	364,921.47	AT5G39660.1
*MpDof06*	Mpo1G32290	Chr1.1: 45817030-45817836: +	268	−0.865	8.76	29.47	249,549.77	AT3G61850.4
*MpDof07*	Mpo1G29540	Chr1.1: 50547300-50547953: +	217	−0.853	8.05	23.59	202,271.63	AT1G51700.1
*MpDof08*	Mpo2G2730	Chr2.1: 8158099-8158155: +	363	−0.657	9.61	38.92	337,721.50	AT3G55370.1
*MpDof09*	Mpo2G35080	Chr2.1: 28281501-28282337: −	289	−0.841	7.67	32.21	269,054.23	AT2G28510.1
*MpDof10*	Mpo2G34260	Chr2.1: 29860047-29860100: +	332	−0.765	9.60	36.38	307,822.14	AT3G55370.1
*MpDof11*	Mpo2G27320	Chr2.1: 38774383-38774523: +	465	−0.632	7.20	50.66	431,187.36	AT3G47500.1
*MpDof12*	Mpo2G24260	Chr2.1: 42219587-42219643: +	377	−0.790	8.33	40.96	350,193.07	AT5G65590.1
*MpDof13*	Mpo2G23730	Chr2.1: 42814492-42815355: +	287	−0.791	8.38	32.19	266,526.77	AT1G28310.2
*MpDof14*	Mpo3G1810	Chr3.1: 2401530-2401697: +	483	−0.976	6.64	54.01	448,411.65	AT5G39660.1
*MpDof15*	Mpo3G5380	Chr3.1: 8429491-8430576: −	379	−0.724	9.06	41.04	352,447.53	AT3G55370.2
*MpDof16*	Mpo3G8630	Chr3.1: 22387948-22388001: +	349	−0.801	9.50	37.60	323,816.27	AT3G55370.1
*MpDof17*	Mpo3G16790	Chr3.1: 35965767-35965865: +	426	−0.742	6.98	46.80	396,324.71	AT5G39660.1
*MpDof18*	Mpo3G50430	Chr3.1: 72788559-72789509: −	336	−0.733	7.17	37.24	313,075.94	AT5G62940.1
*MpDof19*	Mpo3G44960	Chr3.1: 78940187-78941191: −	334	−0.831	4.46	37.50	310,811.41	AT3G52440.1
*MpDof20*	Mpo3G44270	Chr3.1: 79688172-79688195: +	303	−0.868	8.67	33.31	282,187.94	AT3G61850.4
*MpDof21*	Mpo4G36990	Chr4.1: 30377373-30378218: +	281	−0.718	6.83	31.91	261,086.23	AT3G52440.1
*MpDof22*	Mpo4G30010	Chr4.1: 41059335-41059358: +	271	−0.728	7.44	30.08	252,867.00	AT4G24060.1
*MpDof23*	Mpo5G11780	Chr5.1: 15229952-15230992: +	346	−0.905	9.47	37.66	321,294.41	AT5G65590.1
*MpDof24*	Mpo5G15010	Chr5.1: 20897867-20898073: +	383	−0.722	7.54	42.10	355,269.3	AT5G62430.1
*MpDof25*	Mpo5G15000	Chr5.1: 20907162-20907368: +	371	−0.758	7.75	41.06	344,143.11	AT5G62430.1
*MpDof26*	Mpo5G19310	Chr5.1: 39023975-39025201: −	471	−0.987	5.39	52.22	438,204.34	AT3G47500.1
*MpDof27*	Mpo5G25900	Chr5.1: 45067640-45068944: −	495	−0.779	5.07	54.15	459,584.49	AT5G39660.1
*MpDof28*	Mpo6G12720	Chr6.1: 4264598-4265392: +	264	−0.846	8.36	29.71	245,622.01	AT5G66940.1
*MpDof29*	Mpo6G12400	Chr6.1: 4662282-4663304: +	340	−0.449	9.83	34.86	315,965.02	AT5G60850.1
*MpDof30*	Mpo6G12390	Chr6.1: 4682171-4683061: −	317	−0.720	8.45	34.93	294,262.61	AT1G28310.2
*MpDof31*	Mpo6G11570	Chr6.1: 5683475-5683948: −	157	−0.841	8.58	17.66	147,333.83	AT1G29160.1
*MpDof32*	Mpo6G0728L	Chr6.1: 32599558-32600103: +	181	−0.919	9.21	20.62	167,666.87	AT5G60850.1
*MpDof33*	Mpo6G21710	Chr6.1: 49927718-49927744: +	292	−0.934	7.64	32.48	272,496.68	AT2G28510.1
*MpDof34*	Mpo6G20540	Chr6.1: 51515673-51515723: +	302	−0.762	9.55	33.42	280,022.16	AT3G55370.1
*MpDof35*	Mpo7G21240	Chr7.1: 3813449-3814384: −	339	−0.849	8.12	37.71	315,443.59	AT3G47500.1
*MpDof36*	Mpo7G21250	Chr7.1: 3820233-3821276: −	383	−0.700	8.32	41.99	356,634.29	AT3G47500.1

^1^ Grand average of hydropathy.

**Table 2 plants-12-01831-t002:** Ka/Ks analysis and estimated duplication time for the duplicated *MpDof* gene pairs.

Gene Pairs	Duplication Type	Ka ^1^	Ks ^2^	Ka/Ks	Duplication Time/(Million Years)
*MpDof01* and *MpDof32*	Segmental	0.322414	1.063264	0.30323	35.44
*MpDof04* and *MpDof23*	Segmental	0.302491	0.936352	0.323053	31.21
*MpDof06* and *MpDof20*	Segmental	0.354475	0.786864	0.450491	26.23
*MpDof13* and *MpDof28*	Segmental	0.192986	0.643507	0.299897	21.45
*MpDof12* and *MpDof30*	Segmental	0.284244	0.691025	0.411337	23.03
*MpDof11* and *MpDof14*	Segmental	0.426891	2.090931	0.204163	69.70
*MpDof11* and *MpDof24*	Segmental	0.323181	1.270375	0.254398	42.35
*MpDof11* and *MpDof27*	Segmental	0.420527	1.795379	0.234228	59.85
*MpDof10* and *MpDof34*	Segmental	0.329122	0.759796	0.433171	25.33
*MpDof09* and *MpDof33*	Segmental	0.187997	0.777575	0.241774	25.92
*MpDof17* and *MpDof35*	Segmental	0.309166	1.261554	0.245068	42.05
*MpDof14* and *MpDof25*	Segmental	0.40667	2.800724	0.145202	93.36
*MpDof14* and *MpDof27*	Segmental	0.251136	1.038649	0.241791	34.62
*MpDof24* and *MpDof25*	Tandem	0.033197	0.053723	0.617927	1.79
*MpDof35* and *MpDof36*	Tandem	0.102767	0.293985	0.349567	9.80

^1^ Nonsynonymous substitution rate. ^2^ Synonymous substitution rate.

## Data Availability

Not applicable.

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
