# Peer review of "Genome-Wide Identification and Expression Analysis of the Dof Transcription Factor in Annual Alfalfa Medicago polymorpha"

_plants, 2023, doi:10.3390/plants12091831_

Round 1
Reviewer 1 Report
In this long, rather unfocused manuscript, the authors report an inventory of genes encoding Dof transcription factors in alfalfa and describe their in silico analyses of those genes, as well as qPCR analysis of selected Dof genes. The manuscript identified the Dof gene family, and it is a good start for further functional characterizations of these genes. However, too many results or conclusions in this work were based on bioinformatics and prediction, which hurt the quality of the manuscript. In addition, the manuscript is not well written, and a few grammatical mistakes and errors need to be corrected.
1) As mentioned in the discussion part, similar work had been done on its relatives, M. truncatula and M. sativa. Thus, one of the important questions is to compare the difference of Medicago Dof genes between the authors’ and other findings.
2) The only laboratory job is the validation of MpDof expressions under different environmental stresses. Only a limited number of MpDof genes were selected for qRT-PCR analysis. Were they randomly selected from the gene list?
3) The discussion is too long, and some paragraphs are simply descriptions or duplications of results. Please rewrite this part, with a focus on the novel findings from your investigations.
4) Some descriptions sound a little weird to me. For example, in the abstract, the author mentioned “evolutionary stress”. Could you explain the concept to me? In Line 309, “substance metabolism”? Please check your manuscript, and use the scientific descriptions.
Reviewer 2 Report
In this manuscript, the authors conducted genome-wide identification, evolutionary, and expression analysis of the Dof transcription gene family in M. polymorpha. The expression levels of MpDof10, MpDof25, MpDof26, and MpDof29 were obviously up-regulated under drought, salt, and cold stress.
Major comments:
1. What is the basis for the different conditions of dealing with adversity?
2. What is the reason for your choice? Details are required.
Minor comments:
1. Figures have a low resolution and is not clear in this paper.
2. Lines 524, 542, 562 Add the references.
3. The reference format needs to be carefully modified.
Reviewer 3 Report
INTRO. No information on the evolution of DOF in plants is provided. What is current knowledge on the evolution in terms of their gene structure and function? Considering the fact that Cr contains only 1 DOF, it is understandable that this gene family has gone through a lot of changes during evolution, which if available should be given in intro and if not available, should be included in the MM and Results section.
INTRO. No hypothesis or problem stated. What is missing in terms of this gene family in Mp? Why did you choose to study this gene family? Just because no one else did so is not enough? What gap will this study fill? Without such logic, how this study tells any novel information.
RESULTS. The R section immediately starts with a table, which is quite strange. I suggest insert the table at least after the text which cites this table.
RESULTS. Similar to table 1, the authors presented fig 1. before it was cited.
RESULTS. It is good that there are three other species included however, this is not sufficient from studying the evolution point of view. Including one species from each major lineage in kingdom plantae would be minimum requirement to understand the evolution. Like, embryophytes, tracheophyte, pinophyte, monocots and dicots. For reference, I will suggest the following or similar articles. https://doi.org/10.1007/s10528-018-9888-z
RESULTS. L180-onwards. The MYA timeline of the duplicated genes should also be presented in comparison to WGD events in the major plantae lineages suggested in above comment. It will be logical to draw conclusion if the WGD was the main force in expansion of this gene family or not.
RESULTS. In figures, the DOF names should be italicized as done in the text.
DISCUSSION. The D section is weak, it do not reflect about the following.
a. How the gene structure in this species is different from the others.
b. How the differences/similarities in gene structure are relevant to function.
c. What happened during evolution of this gene family in plants?
d. What gap are these results filling.
e. Was there a kind of neofunctionalization?
f. I don’t understand without the above points, why such a long D section is presented. Especially from the novelty point of view, there is limited discussion done.
Round 2
Reviewer 1 Report
OK for publication.
Reviewer 2 Report
The author has made detailed and serious modifications, and it is recommended to agree to accept.
Reviewer 3 Report
the authors write contrasting statements in intro l93-95 that "the function of most genes is not clear, especially the ...." and in next line they say that many angiosperm Dof genes have been identifictid and characterized. Also, they don't provide any reference here about the identified genes.
RESULTS. The R section immediately starts with a table, which is quite strange. I suggest insert the table at least after the text which cites this table. This comment has not been addressed or the authors misunderstood. It is not usual to put the table itself as a first line in the Results.
RESULTS. Similar to table 1, the authors presented fig 1. before it was cited. Same as above.
since authors didn't choose to include other species from evolutionary perspective, they should include this explanation/reason either in the last paragraph of introduction or in the respective section of the results. "the analysis of dicotyledonous model plant Arabidopsis, monocotyledonous model plant rice, alfalfa model plant M. truncatula, and M. polymorpha can provide a basis for studying the evolution of Dof gene in dicotyledonous plants"
: DISCUSSION. Was there a kind of neofunctionalization? this comment was not answered or was misunderstood. Based on the known function of genes present in same cald with same structure or slightly different structure, it can be predicted that if they have similar or novel function in the studied species as compared to the other species used here.
